# Direct observation of core-shell structure and water uptake of individual submicron urban aerosol particles

Ruiqi Man[1], Yishu Zhu[1,†], Zhijun Wu[1,2,*], Peter Aaron Alpert[3,‡], Bingbing Wang[4], Jing Dou[5,§], Jie Chen[5], Yan Zheng[1], Yanli Ge[1], Qi Chen[1], Shiyi Chen[1], Xiangrui Kong[6], Markus Ammann[3], Min Hu[1]

[1]State Key Joint Laboratory of Environmental Simulation and Pollution Control, College of Environmental Sciences and Engineering, Peking University, Beijing, 100871, China

[2]Collaborative Innovation Center of Atmospheric Environment and Equipment Technology, Nanjing University of Information Science and Technology, Nanjing, 210044, China

[3]Laboratory of Atmospheric Chemistry, PSI Center for Energy and Environmental Sciences, Paul Scherrer Institute, Villigen, 5234, Switzerland

[4]College of Ocean and Earth Sciences, State Key Laboratory of Marine Environmental Science, Xiamen University, Xiamen, 361102, China

[5]Institute for Atmospheric and Climate Science, ETH Zürich, Zürich, 8092, Switzerland

[6]Department of Chemistry and Molecular Biology, University of Gothenburg, Gothenburg, 41390, Sweden

[†]now at Department of Earth and Planetary Science, University of California Berkeley, Berkeley, CA, 94720, USA

[‡]now at XRnanotech Gmbh, Parkstrasse 1, Villigen, 5234, Switzerland

[§]now at Institute for Atmospheric and Earth System Research, University of Helsinki, Helsinki, 00014, Finland

[*]*Correspondence to:* Zhijun Wu (zhijunwu@pku.edu.cn)

**Abstract.** Determining the particle chemical morphology is crucial for unraveling reactive uptake in atmospheric multiphase and heterogeneous chemistry. However, it remains challenging due to the complexity and inhomogeneity of aerosols particles. Using a scanning transmission X-ray microscopy (STXM) coupled with near-edge X-ray absorption fine structure (NEXAFS) spectroscopy and an environmental cell, we imaged and quantified the chemical morphology and hygroscopic behavior of individual submicron urban aerosol particles. Results show that internally mixed particles composed of organic carbon and inorganic matter (OCIn) dominated the particle population (73.1 ± 7.4%). At 86% relative humidity, 41.6% of the particles took up water, with OCIn particles constituting 76.8% of these hygroscopic particles. Most particles exhibited a core-shell structure under both dry and humid conditions, with an inorganic core and an organic shell. Our findings provide direct observational

33 evidence of the core-shell structure and water uptake behavior of typical urban aerosols, which

34 underscore the importance of incorporating the core-shell structure into models for predicting the

35 reactive uptake coefficient of heterogeneous reactions.

36 **Short summary:** The particle chemical morphology is important to atmospheric multiphase and

37 heterogeneous chemistry. This work directly observed the core-shell structure and water uptake

38 behavior of individual submicron aerosol particles at an urban site and elucidated the potential impact

39 on particle reactive uptake and heterogeneous reactions.

40 **Keywords:** urban air pollution; individual particles; chemical morphology; core-shell structure; water

41 uptake

42 **Table of Contents Graphic:**

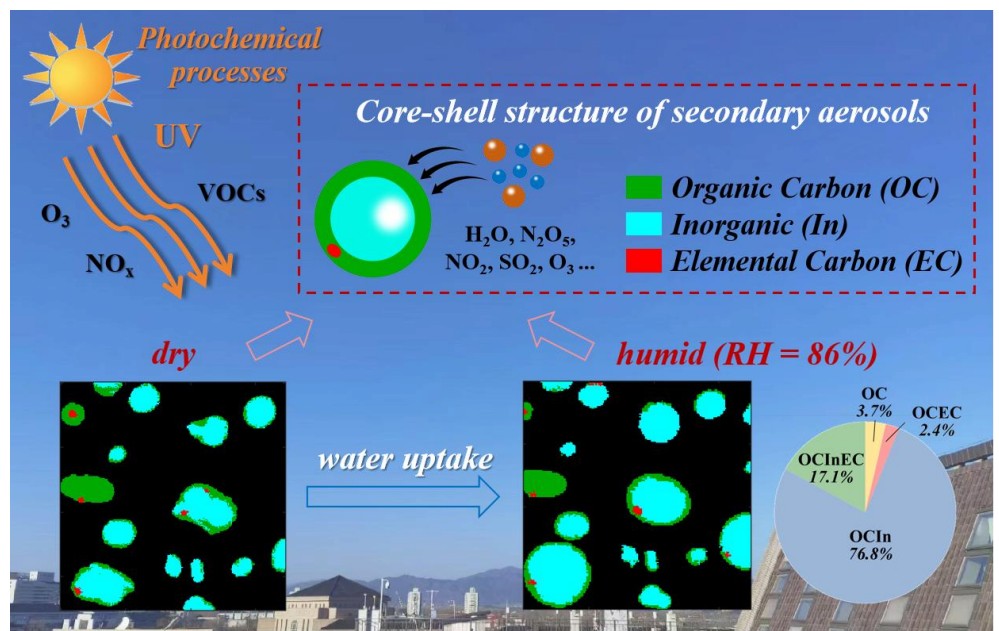

43

## 1 Introduction

Aerosols have significant impacts on visibility, climate, and human health (Mccormick and Ludwig, 1967; Noll et al., 1968; Chow et al., 2006; Rasool and Schneider, 1971). As particles in the atmosphere usually act as reaction vessels for various reactions, the physicochemical properties of aerosol particles play an important role in reactive uptake of gaseous molecules onto particles, mass transfer and gas-particle partitioning equilibrium, and transformation mechanisms of pollutants (Abbatt et al., 2012; Davidovits et al., 2011; George et al., 2015; George and Abbatt, 2010; Su et al., 2020; Ziemann and Atkinson, 2012). Therefore, quantitatively characterizing the aerosol physicochemical properties is vital to atmospheric multiphase and heterogeneous chemistry (Freedman, 2017; Li et al., 2016; Riemer et al., 2019; Tang et al., 2016).

The aerosol physicochemical properties, such as the particle size, chemical morphology (defined as the spatial distribution of various chemical components within a particle herein), mixing state, and hygroscopicity vary under different ambient conditions. These properties and their variations have a critical influence on reactive uptake, a key process in multiphase chemistry which is initiated by the collision of a gas-phase reactant with a condensed-phase surface (Davies and Wilson, 2018; Reynolds and Wilson, 2025). Organic coatings of particles with core-shell structures inhibit reactive uptake of dinitrogen pentoxide ($N_2O_5$) by particulate matter by means of affecting mass accommodation, the availability of water for hydrolysis, and mass transport (Wagner et al., 2013; Jahl et al., 2021; Jahn et al., 2021). Without considering the core-shell structure, the reactive uptake coefficient of $N_2O_5$ tends to be overestimated from several times to tens of times (Wagner et al., 2013). Therefore, investigating the particle chemical morphology is necessary for accurately quantifying the uptake coefficient and reducing uncertainty in heterogeneous reactions.

So far, extensive research has been conducted on the physicochemical properties of bulk aerosols by various techniques, such as the Humidified Tandem Differential Mobility Analyzer (H-TDMA), Aerosol Mass Spectrometer (AMS), and Soot Particle Aerosol Mass Spectrometer (SP-AMS) (Li et al., 2016; Tang et al., 2019; Riemer et al., 2019). However, the bulk analysis mostly obtains indirect information about the physicochemical properties of particle populations based on assumptions and estimations, which is difficult to directly observe the chemical morphology and mixing state of aerosol particles (Li et al., 2016). This knowledge gap hinders our understanding of the role of aerosol particles

in reactive uptake and heterogeneous processes. As a comparison, individual particle analysis can
provide direct observational evidence about the chemical morphology and mixing state at the
microscopic scale, which is essential for exploring particle hygroscopic and optical properties (Krieger
et al., 2012; Li et al., 2016; Posfar et al., 2010; Wu and Ro, 2020).
Scanning transmission X-ray microscopy combined with near-edge X-ray absorption fine
structure (STXM/NEXAFS) spectroscopy bases on synchrotron radiation technology. It is a robust
technique for obtaining chemical morphology information of numerous individual particles with high
spectral energy resolution and chemical specificity, as it can identify and distinguish various chemical
composition at the single particle level within a particle population (Moffet et al., 2011; Shao et al.,
2022). The soft X-ray energy range of STXM (100 – 2000 eV for STXM versus 50 – 200 keV for
electron microscopy) makes it possible to quantify light elements (such as carbon, nitrogen, and
oxygen) with little beam damage (Moffet et al., 2011). In addition, STXM doesn't require ultrahigh
vacuum conditions (Moffet et al., 2011). In short, STXM/NEXAFS spectroscopy provides an enhanced
chemical sensitivity for obtaining specific organic chemical bonds, functional groups, and speciation
information, which has enormous potential in exploring ambient samples under atmospheric relevant
conditions, especially submicron-sized particles.
Compared with other STXM endstations which generally analyze samples under vacuum
conditions (Alpert et al., 2022; Bondy et al., 2018; Fraund et al., 2020; Knopf et al., 2023; Lata et al.,
2021; Moffet et al., 2010a; Moffet et al., 2010b; Moffet et al., 2013; Moffet et al., 2016; Tomlin et al.,
2022), several STXM instruments are equipped with an in-situ temperature and relative humidity (RH)
control environmental cell, allowing for investigating hygroscopicity and water uptake behavior of
laboratory-generated particles (Ghorai and Tivanski, 2010; O'brien et al., 2015; Piens et al., 2016;
Zelenay et al., 2011a; Zelenay et al., 2011b). However, only a few researches focusing on hygroscopic
behavior of ambient particles has been reported so far, and these particles were collected in rural
environment (Piens et al., 2016) or forest (Mikhailov et al., 2015; Pöhlker et al., 2014). Studies on
water uptake of urban aerosol particles using STXM and corresponding knowledge for their chemical
morphology under humid conditions is currently lacking.
In recent years, the air quality in China has improved notably due to the implementation of a
series of strict pollution mitigation measures. These improvements are attributed to decreasing primary
emissions, while the contributions of secondary species to particle mass have become more significant
(Lei et al., 2021; Wang et al., 2019). To elucidate the causes and mechanisms of pollution episodes in
China, numerous research has been carried out on the pollution characteristics (Gao et al., 2015; Gao et
al., 2018; Guo et al., 2014; Huang et al., 2014; Liu et al., 2018; Sun et al., 2013; Wang et al., 2014;
Zhao et al., 2013) and physicochemical properties (Gao and Anderson, 2001; Li et al., 2017; Shen et al.,
2019; Song et al., 2022) of ambient aerosols. However, there is still a lack of study on direct
observation of the chemical morphology and hygroscopic behavior of secondary urban aerosols at the
single particle level. This knowledge gap hinders our understanding of the role of secondary aerosols as
reaction vessels in heterogeneous reactions.
In this study, we investigated the chemical morphology of ambient individual submicron aerosol
particles using STXM/NEXAFS spectroscopy. The ambient samples were collected at an urban site in
North China Plain, Beijing during a pollution episode. We also explored the chemical morphology and
water uptake behavior of individual particles at high humidity (RH = 86%) using an environmental cell.
This work aims to improve our comprehension of the physicochemical properties of particles in typical
urban pollution atmospheres, aiding in clarifying their atmospheric heterogeneous processes and
multiphase chemistry.
**2 Materials and methods**
**2.1 Sampling and instruments**
To study the physicochemical properties of ambient particles, samples were collected during a
pollution episode at the Peking University Urban Atmosphere Environment Monitoring Station
(PKUERS, 39°59′21″N, 116°18′25″E) in Beijing, China. More details about the measurement site can
be found in our previous studies (Tang et al., 2021; Wu et al., 2007).
The individual particle sample was collected using a four-stage cascade impactor with a Leland
Legacy personal sample pump (Sioutas, SKC, Inc., the US) at a flow rate of 9 L min$^{-1}$. The sampling
started at 5:04 P.M. on October 1$^{st}$, 2019 and lasted for 5 minutes. The sampling substrate was a copper
grid (Lacey Carbon 200 mesh, Ted Pella, Inc., the US) suitable for its X-ray transparency. Particles
collected onto the last stage with the 50% cut-point aerodynamic diameter of 250-nm were used for
STXM analysis. The sample was placed into a sample box sealed with a bag filled with nitrogen, and it
was stored in a freezer at a temperature of –18°C until analysis. Previous results indicate that the

chemical composition of organic aerosols (especially secondary organic aerosols, SOA) and the mass concentrations of black carbon (BC) both remained stable for several weeks under low-temperature storage conditions (–20℃ and –80℃ for organic aerosols, and 2℃, 4℃, and 5℃ for BC); while significant changes occurred over time when samples were stored at room temperature even just for a few days (Mori et al., 2016; Mori et al., 2019; Resch et al., 2023; Ueda et al., 2025; Wendl et al., 2014).

Other parameters were measured from September 28th to October 7th, 2019. The non-refractory chemical composition of submicron particles (NR-PM$_1$) was obtained by a Long Time-of-Flight Aerosol Mass Spectrometer (LTOF-AMS, Aerodyne Research Inc., the US) (Zheng et al., 2020; Zheng et al., 2023). Calibrations of ionization efficiency (IE) and relative IE followed the standard procedures described in previous studies (Canagaratna et al., 2007; Fröhlich et al., 2013). The reference temperature and pressure conditions of mass concentrations reported herein were 293.7 K and 101.82 KPa. We applied composition-dependent collection efficiency (CDCE) values (0.50 ± 0.01, mean ± standard deviation) that were calculated by the methods introduced by Middlebrook et al. (2012) to the AMS data. The mass concentration of fine particles (PM$_{2.5}$) was measured by a TEOM analyzer (TH-2000Z1, Wuhan Tianhong Environmental Protection Industry Co., Ltd., China). Meteorological parameters including temperature ($T$), RH, wind speed, and wind direction were monitored by an integrated 5-parameter Weather Station (MSO, Met One Instruments, Inc., the US).

**2.2 STXM/NEXAFS analysis**

In order to gain the chemical morphology, mixing state, and component information of individual particles, STXM/NEXAFS spectroscopy measurements were carried out at the PolLux beamline (X07DA) of the Swiss Light Source (SLS) at Paul Scherrer Institute (PSI) (Raabe et al., 2008). In brief, X-rays illuminated a Fresnel zone plate focusing the beam to a pixel of $35 \times 35$ nm$^2$. The zone plate has a central stop that acts together with another optic known as an order sorting aperture to eliminate unfocused and higher-order light, ensuring only first-order focused light is transmitted to the sample. Then, X-rays transmitted through the sample are detected. The absorbance of each pixel is characterized by optical density (OD) based on the Beer-Lambert's law as follows,

$$OD = -\ln(I/I_0) \tag{1}$$

where $I$ and $I_0$ are the intensity of photons transmitted through a sample region and a sample-free
region, respectively. Further details including the uncertainty estimation of OD are described in the
Supplementary Information (SI).
STXM/NEXAFS spectroscopy scans X-ray energies over particles with high spectral energy
resolution. When inner shell electrons of atoms absorb X-ray photons, they can transition into
unoccupied valence orbitals, resulting in an absorption peak that is used to identify specific bonding
characteristics. The amount of absorption depends on the photon energy ($E$), elemental composition, as
well as sample thickness and density (Moffet et al., 2011). We employed two measurement strategies
to optimize photon flux to the particles, achieving the best signal-to-noise ratio while minimizing the
scan time. The first strategy was a high energy-resolution mode with an X-ray energy resolution $\Delta E =$
0.2 eV and a coarse pixel size of around $100 \times 100$ nm$^2$ to measure absorption at small energy steps.
The energy resolution is defined as being able to distinguish between two absorption peaks separated
by $\Delta E$ at the full width at half maximum OD. In this mode, carbon (C), nitrogen (N), and oxygen (O)
K-edge spectra of individual particles were measured. The energy offset of C and O spectra were +0.4
eV and +1.2 eV respectively, according to the energy calibration procedures using polystyrene spheres
and gas-phase carbon dioxide ($CO_2$). The energy offset of N at the K-edge was not calibrated, however,
the obtained spectra of ambient particles appeared identical to ammonium salts in literature (Ekimova
et al., 2017; Latham et al., 2017). Due to the presence of ammonium, which was confirmed in particles
using AMS, we applied a calibration factor of +0.1 eV for the N K-edge to match our observed main
peak to that of ammonium at 405.7 eV. Optical density detected over the same spot at different photon
energies at the carbon and nitrogen K-edges was displayed in Fig. S1, and less beam damage during the
experiment was confirmed.
The second strategy is a high spatial-resolution mode with a pixel size of $35 \times 35$ nm$^2$ and $\Delta E =$
0.6 eV, where imaged at four specific energies for the C K-edge, namely, 278.0 eV, 285.4 eV, 288.6
eV, and 320.0 eV. Automated analysis followed the methodology of Moffet et al. (2010a) and Moffet
et al. (2016). In brief, absorption at 278.0 eV ($OD_{278.0eV}$) is regarded as the pre-edge of carbon, which is
mainly due to off-resonance absorption by inorganic elements other than carbon. Absorption at 285.4
eV ($OD_{285.4eV}$) is due to the characteristic transition of sp$^2$ hybridized carbon (i.e., doubly bonded
carbon). Since this peak is abundant for elemental carbon (EC), it can be used to discern soot, because
EC is a type of components of soot (Penner and Novakov, 1996). Absorption at 288.5 eV ($OD_{288.5eV}$)
comes from carboxylic carbonyl groups, which are common in organic aerosols in atmospheres.
Therefore, organic carbon (OC) is identified by this energy. Absorption of the post-edge at 320.0 eV
($OD_{320.0eV}$) is contributed by carbonaceous and non-carbonaceous atoms (Moffet et al., 2010a).
Based on absorption at these four typical energies, we obtain three images by further processing.
The difference between OD at the post-edge and OD at the pre-edge ($OD_{320.0eV}$ - $OD_{278.0eV}$) indicates
total carbon. The ratio of OD at the pre-edge to OD at the post-edge ($OD_{278.0eV}$ / $OD_{320.0eV}$) indicates the
relative absorption contribution of inorganic matter (In). Compared with the absorbance contribution of
doubly bonded carbon to total carbon ($\%sp^2$) in the highly oriented polycrystalline graphite (HOPG,
assuming that $\%sp^2 = 100\%$) at 285.4 eV, the spatial distribution of EC/soot in samples can be
identified by the procedure of Hopkins et al. (2007). It is assumed that total carbon consists of OC and
EC. The thresholds of these images follow the criteria mentioned in Moffet et al. (2010a) and Moffet et
al. (2016). These three images described above were then overlaid to create a chemical map of
individual particles.
**2.3 Criterion of particle water uptake based on the total oxygen absorbance**
To determine whether particles took up water, a criterion was established on the basis of the total
oxygen absorbance determined at the energy of 525.0 eV (the pre-edge of oxygen) and 550.0 eV (the
post-edge of oxygen). Based on the same principle as the total carbon calculation, the difference
between OD at the post-edge and pre-edge of oxygen represents the total oxygen absorbance. Due to
the fact that each particle is composed of some pixels, the total oxygen absorbance ($\Delta OD$) of an
individual particle under dry and humid conditions is calculated as follows,

$$\Delta OD_{dry} = \sum_{i=1}^{m} \Delta OD_i = \sum_{i=1}^{m} (OD_{post,i} - OD_{pre,i}) = \sum_{i=1}^{m} OD_{post,i} - \sum_{i=1}^{m} OD_{pre,i} \qquad (2)$$

$$\Delta OD_{humid} = \sum_{j=1}^{n} \Delta OD_j = \sum_{j=1}^{n} (OD_{post,j} - OD_{pre,j}) = \sum_{j=1}^{n} OD_{post,j} - \sum_{j=1}^{n} OD_{pre,j} \qquad (3)$$

where $m$ and $n$ are numbers of pixels that make up an individual particle under dry and humid
conditions, $i$ and $j$ are a certain pixel within an individual particle under dry and humid conditions, *post*
and *pre* respectively represent the energy at the post-edge (550.0 eV) and that at the pre-edge (525.0 eV)
of oxygen. If a particle takes up water, the amount of oxygen atoms within this particle will increase,
leading to an amplification in $\Delta OD$. Water uptake may increase particle height and absorption. On the

other hand, it possibly causes a particle to spread out, which may reduce particle height and thus absorption. Although a thinner particle that contains more water may result in less absorption at some specific pixels, $\Delta OD_{humid}$ will be larger than $\Delta OD_{dry}$ due to the fact that more pixels are summed, i.e., $n > m$. Therefore, comparing the results of Eq. 2 and Eq. 3 will quantify the total oxygen absorbance of a particle under dry and humid conditions, and determine particle water uptake. Specifically, if $\Delta OD_{humid} > \Delta OD_{dry}$, then we assume that a particle has taken up water.

**2.4 A novel in-situ environmental cell**

To explore the chemical morphology and hygroscopicity of the particles under humid conditions, we adjusted the RH of an in-situ environmental cell with sample placed in it. The environmental cell can also be used for trace gas reactive uptake and photochemical reactions with laboratory-generated particles (Alpert et al., 2019; Alpert et al., 2021). The environmental cell used in this study consists of a removable sample clip that hosts a sealed silicon nitride (SiNit) window and a main body that contains gas supply lines and temperature control. Together, they are mounted in the STXM vacuum chamber. A SiNit window at the back side of the main body is also sealed and ensures X-ray transparency passing through the whole environmental cell assembly. Descriptions of the connections for the gas supply, heating and cooling devices, and temperature measurement can be found in previous studies (Huthwelker et al., 2010; Zelenay et al., 2011a). The detailed methods of collecting the ambient particles by the impactor and measuring them in the environmental cell were shown in the SI.

We performed humidity calibration experiments to make sure sufficient heat transfer and a homogeneous water vapor field across the samples. It is important due to the fact that the only way for samples to gain or lose heat and water was through air contact. To study the accuracy of RH in the environmental cell, water uptake and deliquescence of a sodium chloride (NaCl) standard sample was observed. The deliquescence relative humidity (DRH) of pure NaCl crystals obtained from literature and thermodynamic models is around 75 – 76% at room temperature (Eom et al., 2014; Martin, 2000; Peng et al., 2022). The images of the NaCl sample displayed in Fig. S2 illustrate the morphological changes as RH increased. As shown in Fig. S2, particle morphologies in panels (A) – (C) remained essentially identical before RH reached the DRH of NaCl, although the focus position slightly varied in different panels. When RH was 75.6% (Fig. S2D), particles completely deliquesced and some coalesced. The uncertainty of RH in the environmental cell in this study was determined conservatively

to be ±2%, in agreement with previous results (Huthwelker et al., 2010). Information about the oxygen
K-edge spectra of the NaCl sample at high RH can be found in Fig. S3.

**3 Results and discussion**

**3.1 Pollution characteristics during the sampling period**

Time series of meteorological parameters, mass concentrations of gaseous pollutants, PM$_{2.5}$, and
NR-PM$_1$ are shown in Fig. 1. During the pollution episode from September 29[th] to October 3[rd], 2019,
the stagnant weather condition with low wind speed led to pollution accumulation. The air became
clean due to the appearance of a strong north wind on October 4[th] (Fig. 1A). The sampling time of the
individual particle sample was 5:04 P.M. on October 1[st] (see the red line in Fig. 1) with an ozone (O$_3$)
concentration of 97.1 ppb. The average mass fractions of chemical composition of NR-PM$_1$ during the
sampling period of individual particles could be found in Fig. S4. During this period, the low mass
fraction of volatile inorganic species such as nitrate made it suitable for measurements using offline
techniques, such as STXM, because the loss of volatile species during storage and measurement
processes was minimal.
During the pollution episode, the maximum daily 8-hour average of ozone (MDA8-O$_3$) was 110.3
± 10.1 ppb (i.e., 236.5 ± 21.7 μg m$^{-3}$). The concentration of O$_x$ [O$_x$ = nitrogen dioxide (NO$_2$) + O$_3$] was
88.6 ± 29.4 ppb (Fig. 1B), reflecting a high atmospheric oxidation capacity that drives secondary
transformations of gaseous pollutants (Dou et al., 2024; Xiao et al., 2022). The average PM$_{2.5}$ was 74.3
± 18.3 μg m$^{-3}$ (Fig. 1C). As shown in Fig. 1D – 1E, the average mass concentration of secondary
inorganic aerosol (SIA) in NR-PM$_1$ was 21.3 ± 4.8 μg m$^{-3}$, with sulfate and nitrate contributing almost
equally to particle mass (i.e., 18.1% and 17.3% respectively). Organic matter in NR-PM$_1$ had an
average mass fraction of 56.4% (Fig. 1E). The mass concentrations of primary organic aerosol (POA)
and SOA were estimated based on the positive matrix factorization (PMF) analysis (Ulbrich et al.,
2009). As shown in Fig. 1F, SOA dominated organic matter, contributing an average of 68.9%. Overall,
this pollution episode was led by secondary oxidation processes and featured by high contributions of
secondary particulate species.

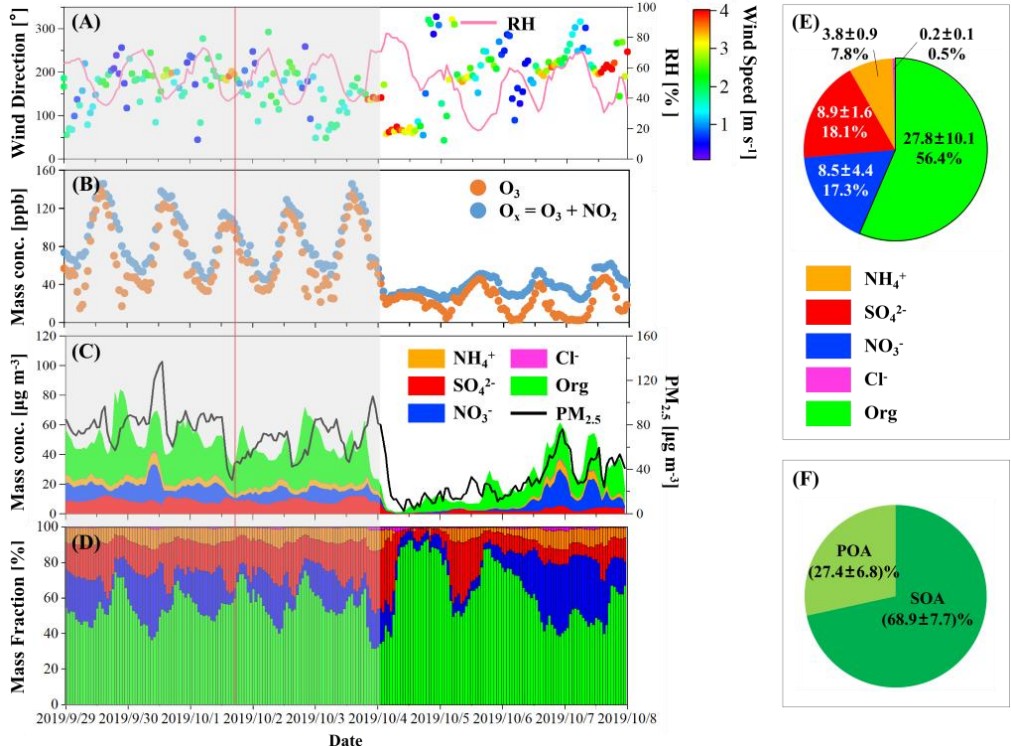

Figure 1: Time series of (A) wind direction, wind speed, and relative humidity (RH), (B) mass concentrations of ozone ($O_3$) and $O_x$ ($O_3$ + nitrogen dioxide, $NO_2$), (C) mass concentrations of fine particles ($PM_{2.5}$) and non-refractory submicron particles (NR-$PM_1$), and (D) mass fractions of chemical composition of NR-$PM_1$ are shown. The gray area represents the pollution episode lasting from September 29th to October 3rd. The red line indicates the sampling time for the individual particle sample. (E) Pie chart showing the average mass fractions of chemical composition of NR-$PM_1$ during the pollution period. The number in the first row of each part is the average mass concentration and standard deviation (SD) with a unit of μg m⁻³. The number in the second row is the average mass fraction. (F) Pie chart showing the average mass contributions of primary organic aerosol (POA, light green) and secondary organic aerosol (SOA, dark green) to the total organic. Average mass fraction and SD are marked in the pie chart.

**3.2 Chemical maps of individual particles**

Chemical maps of individual particles under dry conditions are displayed in Fig. 2. Different images denote particles located in different regions of interest (ROI, 12 in total) of the sampling substrate. 197 individual particles were investigated in total. The detailed images of total carbon, inorganic, and doubly bonded carbon maps are available in Fig. S5 – S7. Most submicron particles on the substrate were round or nearly round, while supermicron particles predominantly exhibited irregular shapes. The circular equivalent diameter of individual particles was calculated, with the methods detailed in the SI. The normalized size distribution of overall particles followed a normal distribution, with a mean diameter ± standard deviation (SD) being 0.83 ± 0.30 μm (Fig. S8A). A significant proportion of the particles were within the 0.4 – 1.2 μm size range.

As displayed in Fig. 2, chemical maps of individual particles showed that they were dominated by
inorganic substances (colored in cyan), which were likely sulfate that was frequently observed by AMS
(Fig. 1C). Approximately one quarter (24.9%) of the particles contained EC/soot (colored in red).
Notably, around 82% of these soot-containing particles had soot located at particle edges. One of the
possible reasons is that inorganic species (such as crystals) pushed soot away from the center of the
particles during their efflorescence (Moffet et al., 2016). While, it should be noted that particle
deformation may occur during the particle collection process due to the high particle impaction
velocity of the impactor (O'Brien et al., 2014). Therefore, the distribution of chemical components
within individual particles displayed in the images may differ from that of aerosol particles in ambient
atmospheres. Additionally, several particles contained multiple soot components, which was also
observed before (Moffet et al., 2016).
Typically, inorganic components and/or soot were encased in organic matter, forming a core-shell
structure characterized by an inorganic-dominated core and an organic-dominated shell. Figure 2
illustrates that most organic-inorganic internally mixed particles exhibited thin coatings, likely from
fresh emissions. Conversely, a few particles have thick coatings, which is indicative of aging processes
in a highly active photochemical environment. Previous studies suggest that most of the
soot-containing particles with thin coatings would have rather smaller absorption enhancement
compared with those with thick coatings (Bond et al., 2006; Moffet et al., 2016).
The observed core-shell morphology could also result from liquid-liquid phase separation (LLPS),
influenced by fluctuating ambient RH (Fig. 1A) and determined by the oxygen-to-carbon (O:C) ratio of
the organic fraction (Freedman, 2020; Li et al., 2021; You et al., 2012; You et al., 2014; Freedman,
2017). To test this hypothesis, the O:C ratio of the individual particles composed of pure organic
composition was estimated as $0.53 \pm 0.15$ based on the STXM data. The estimation methods were
displayed in the SI. This falls within the threshold range for LLPS occurrence in ammonium sulfate -
organic mixing particles ($0 < O:C < 0.57$) (You et al., 2013). For comparison, the value of the O:C ratio
by AMS during the individual particle collection period is also calculated ($0.60 \pm 0.01$), and the data
set was displayed in the SI. The difference between the O:C ratio results by STXM and LToF-AMS
may be due to the reasons as follows: (1) STXM measures individual particles, while AMS targets bulk
aerosols; (2) Particles collected onto the last stage using a four-stage cascade impactor with the 50%
cut-point aerodynamic diameter of 250-nm were used for STXM analysis, while AMS measured the
non-refractory chemical composition of submicron particles.

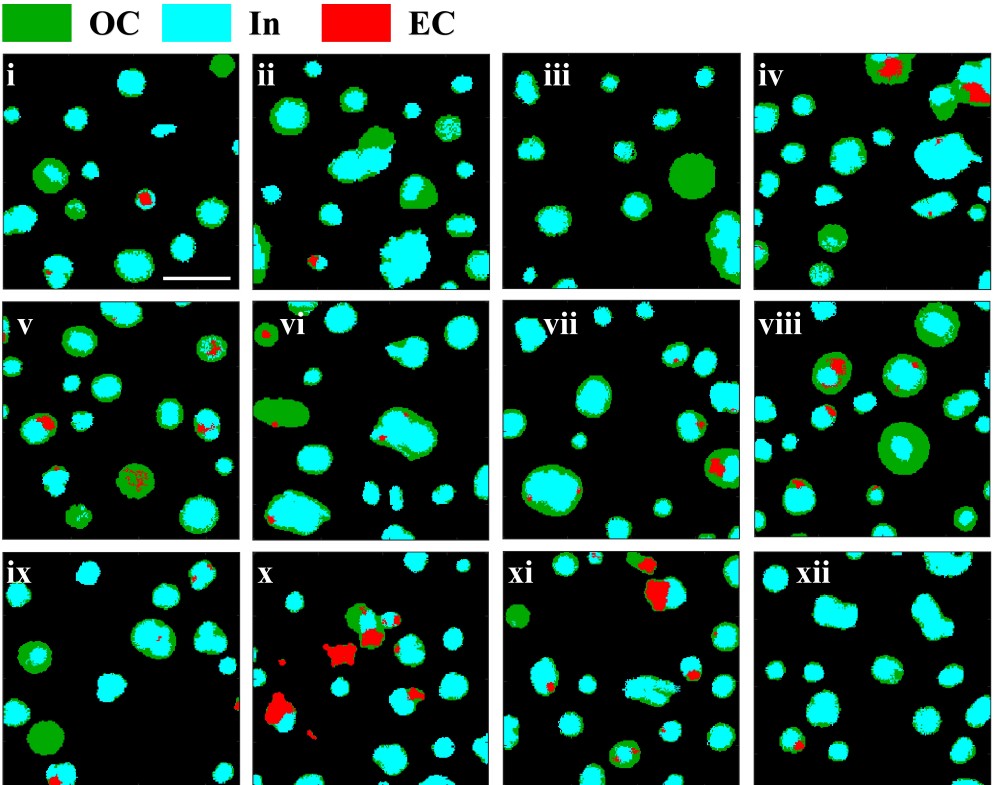


**Figure 2: Chemical maps of individual particles in 12 regions of interest (ROI) of sampling substrate under dry conditions on the basis of pixels. Green, cyan, and red color represent dominant components of organic carbon (OC), inorganic matter (In), and elemental carbon (EC), respectively. The scale bar in the upper left image represents 2 µm and applies to all the images.**

Statistically, the particles were categorized into four types based on their mixing state, including
pure organic (OC), organic internally mixed with soot (OCEC), organic internally mixed with
inorganic (OCIn), and organic internally mixed with inorganic and soot (OCInEC). OCIn particles
were the most abundant type in the examined particle population (73.1 ± 7.4%), followed by OCInEC
(20.8 ± 6.7%) and OCEC (4.1 ± 3.3%), indicating a highly internally mixed particle population. Pure
organic particles only accounted for 2.0 ± 2.3%. The calculation of the margin of error of the mixing
state proportions can be found in the SI. The mean diameters of OCEC, OCIn, and OCInEC particles
were 0.66, 0.79, and 1.02 µm, respectively (Fig. S9). This suggests that the internally mixed particles
containing three species families tend to be larger than those composed of two species families.
**3.3 The effects of particle water uptake on chemical maps**
Chemical maps of individual particles under humid conditions (RH = 86%) measured in the
environmental cell were displayed in Fig. 3, and these ROI are identical and matched one by one to
those in Fig. 2. For comparison, the one-to-one particle chemical maps of the same region of interest
under both dry and humid conditions can be found in Fig. S10. It was observed that many particles
tended to be more rounded due to water uptake at high RH, especially for particles with diameters in
the supermicron range (e.g., particles in (ii), (vi), (xi), and (xii) in Fig. 3). If particles take up
significant amounts of water and are homogeneously mixed, they would appear as dominated by
inorganic (colored in cyan) due to absorption of a large amount of water at the carbon pre-edge. In
contrast, most particles remained inhomogeneous and exhibited a core-shell structure under humid
conditions. A possible reason is that the settled RH may not reach the mixing relative humidity (MRH)
of particles, which is defined as a threshold where different phases in an aqueous particle mix into one
homogeneous phase. This MRH usually varies from 84% to over 90% (Li et al., 2021; You et al., 2014;
Zhang et al., 2022).
Additionally, around 87% of soot was located at the edge of the humidified particles, with no
obvious location change of soot observed in most particles. A previous study witnessed the
redistribution of soot within phase-separated particles only after the phase mixing process occurred
(Zhang et al., 2022), which is consistent with the phenomenon observed in our study.

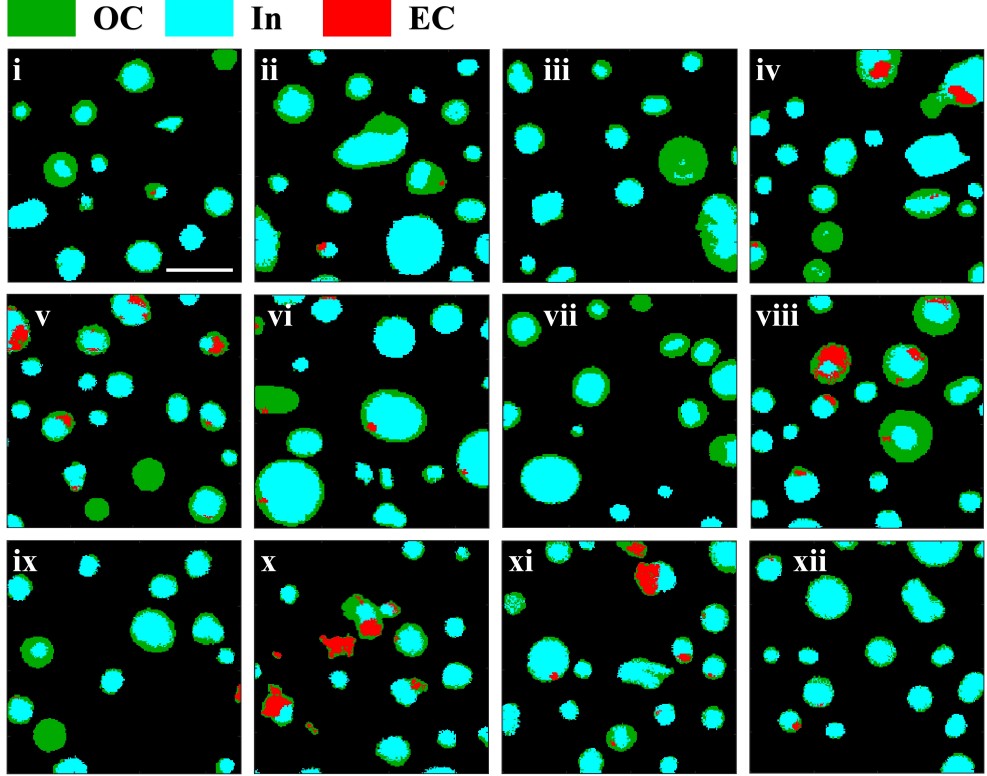


**Figure 3: Chemical maps of individual particles in 12 ROI of sampling substrate under humid conditions (RH = 86%) measured in an in-situ environmental cell. Green, cyan, and red color represent dominant components of OC, In, and EC, respectively. The scale bar in the upper left image represents 2 μm and applied to all the images.**

Comparing size distributions of particle populations under dry (Fig. S8A) and humid conditions (Fig. S8B) reveals that they exhibited similar distribution characteristics. The mean diameter of overall particles at high RH was 0.86 ± 0.33 μm, compared with 0.83 ± 0.30 μm under dry conditions. This indicates that the overall size distribution of the humidified particles shifted a little towards larger particles due to water uptake. Specifically, approximately 56.3% of the particles showed an average increase of 14.9% in diameter, while the remaining exhibited an average decrease of 8.2%. Pöhlker et al. (2014) also observed this abnormal phenomenon where some particles decreased in size with increasing RH. They suggested that it could be attributed to the decreasing viscosity and increasing surface tension due to particle water uptake at high RH. This led to larger contact angles between the collected particles and the substrate, causing the particles to "bead up" and therefore reducing their cross-section areas in the view (Pöhlker et al., 2014). In addition, one should note that a small number of particles at the edge of the ROI did not entirely enter the field of view due to the limited observation range, which may slightly affect the quantification of their size.

According to the criterion for water uptake by individual particles based on the total oxygen absorbance described in the Sect. 2.3, 41.6% of the particles took up water. As shown in Fig. S11A, OCIn particles were the dominant mixing state type taking up water (76.8%), followed by OCInEC (17.1%). There were also several OCEC (2.4%) and OC (3.7%) particles displayed water uptake. Different particle mixing state types exhibited distinct patterns of hygroscopic behavior. For instance, 43.8% of OCIn particles took up water, while 34.1% of OCInEC particles performed the same. This difference may be attributed to varying hygroscopicity of different components. For example, the single hygroscopic parameter (κ) of ammonium nitrate, ammonium sulfate, ammonium hydrogen sulfate, POA, and SOA is 0.58, 0.48, 0.56, 0, and 0.1, respectively (Wu et al., 2016). Based on the AMS data, κ of bulk aerosols during the sampling period (0.25 ± 0.01) was calculated according to κ-Köhler theory (Stokes and Robinson, 1966; Petters and Kreidenweis, 2007), indicating a relatively low hygroscopic capacity of NR-PM$_1$ during sampling, which could explain why only less than half of the particles exhibited water uptake at such high humidity conditions. In addition, the average diameter

of particles taking up water increased from $0.82 \pm 0.33$ µm to $0.91 \pm 0.36$ µm. The relative frequency
distribution and the size-resolved fraction of particles taking up water can be found in Fig. S11B.
**3.4 Chemical composition of ambient submicron particles**
NEXAFS spectra with high energy resolution were measured at the C (278 – 320 eV), N (395 –
430 eV), and O (525 – 550 eV) K-edges. As shown in Fig. 4A, three notable absorption peaks at the C
K-edge were observed at 285.4, 286.7, and 288.6 eV. According to previous literature (Warwick et al.,
1998; Moffet et al., 2010a), the peak at 285.4 eV refers to the characteristic transition of $sp^2$ hybridized
carbon (C $1s \rightarrow \pi^*_{R(C^*=C)R}$). The peak at 286.7 eV may result from the transition of ketonic carbonyl (C
$1s \rightarrow \pi^*_{R(C^*=O)R}$), representing ketone and ketone-like compounds. The peak appearing at 288.6 eV
represents the characteristic transition of carboxylic carbonyl functional groups (C $1s \rightarrow \pi^*_{R(C^*=O)OH}$),
which refers to organic matter and is generally dominant in the outer shell of particles (Moffet et al.,
2016; Prather et al., 2013). One should note that there could be extra components in both core and shell
in a phase-separated particle with an inorganic-rich core and an organic-rich shell, for example, organic
in core or inorganic in shell (Gaikwad et al., 2022). Therefore, the 288.6-eV peak may also be observed
in a particle core with relatively low peak intensity. In addition, two other peaks at 296.8 and 299.6 eV
were present (see spectra (d) and (e) in Fig. 4A), corresponding to the $L_2$- and $L_3$-edges of potassium
(Moffet et al., 2010a). In our sample, potassium may come from biomass burning processes based on a
previous study (Wu et al., 2017).
Nitrogen K-edge spectra in Fig. 4B illustrate that ammonium salts were the main nitrogen species
in the sample. We observed a broad main peak centered at 405.7 eV, which is the feature of ammonium
(Ekimova et al., 2017). A smaller peak was observed at 401.0 eV, which is absorption due to nitrogen
gas ($N_2$) either trapped in the inorganic crystal or formed under X-ray exposure (Latham et al., 2017).
Absorption of nitrate ($NO_3^-$) and nitrite ($NO_2^-$) commonly have narrow peaks at 405.1 eV and 401.7 eV
(Smith et al., 2015), respectively, which were not apparent in our spectra. This is likely because
particulate nitrite is below the detection limit, or its peak is masked by the pronounced absorption of
ammonium ($NH_4^+$). The solid ammonium nitrate and sodium nitrate salts could exhibit a peak at around
415.0 eV (Smith et al., 2015). However, this was not observed in Fig. 4B. Organic compounds
containing nitrogen, such as amino acids, N-heterocyclics, and nitroaromatic compounds, can be
abundant in urban aerosol particles due to combustion sources (Yu et al., 2024). They have a large
variety of possible peak positions, heights, and widths (Leinweber et al., 2007), making the
identification of these compounds difficult. Although a positive identification of specific organic
nitrides cannot be made, we note that amino acids and 5- or 6-ring heterocycles commonly have narrow
peaks at around 401 eV and broad peaks at 405 eV (Leinweber et al., 2007). We expect that organic
nitrides did contribute to the observed N K-edge spectra, although a targeted study on molecular
identification would be necessary to establish further certainty.

Oxygen K-edge spectra in Fig. 4C exhibited a large peak at 536.9 eV, which is a representative

characteristic of sulfate-rich particles (Colberg et al., 2004; Slowik et al., 2011; Mikhailov et al., 2015;
Pöhlker et al., 2014), consistent with the result of AMS. A smaller peak was observed at 532.5 eV,
confirming the presence of ketone, aldehyde, or carboxyl functionalities (Moffet et al.,2011), which
aligns with the results from C K-edge spectra. These compositions tend to take up water under humid
conditions.

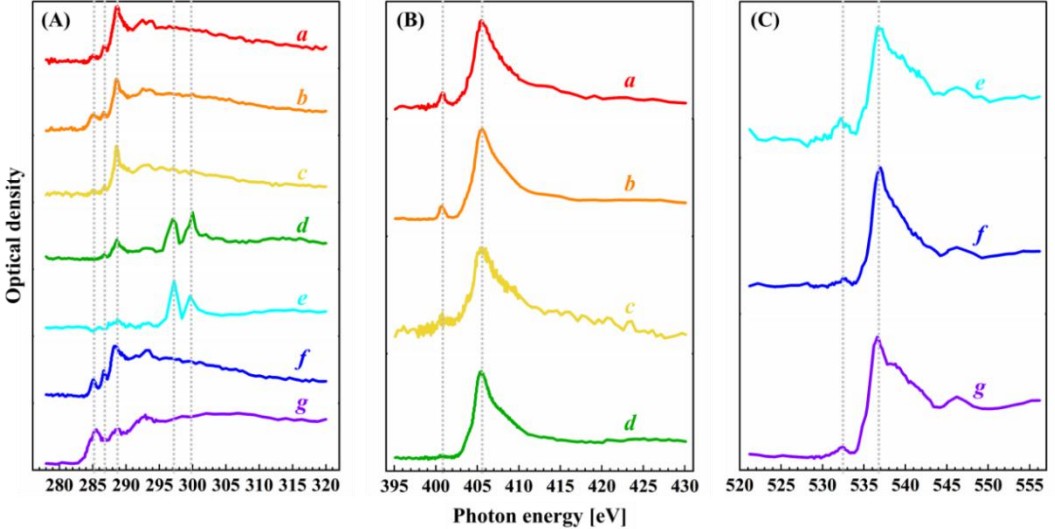


**Figure 4: NEXAFS spectra for individual particles at (A) carbon (C), (B) nitrogen (N), and (C) oxygen (O)**
**K-edges. In panel (A), peaks were observed at 285.4, 286.7, and 288.6 eV, and two typical peaks appeared at**
**around 296.8 and 299.6 eV in spectra (d) and (e). In panel (B), a main peak appeared at 405.7 eV, and a**
**smaller peak appeared at 410.0 eV. In panel (C), a main peak appeared at 536.9 eV, and a smaller peak was**
**at 532.5 eV. Each small case letter of a spectrum stands for the average result of all the pixels within an**
**individual particle. The same letter in different panels doesn't refer to a same particle.**

**4 Conclusions and implications**

Particles in the atmosphere usually act as reaction vessels for heterogeneous reactive uptake of

gaseous molecules, and heterogeneous processes play an important part in gas-particle partitioning and

secondary aerosol formation (Abbatt et al., 2012; Davidovits et al., 2011; Kolb et al., 2010). However, determining the particle physicochemical properties is crucial but challenging due to the complexity and inhomogeneity of aerosols particles (Barbaray et al., 1979; Zong et al., 2022). So far, there is a lack of study on direct observation of the physicochemical properties of urban aerosols at the single particle level under different conditions, which hinders our understanding of the role of urban particle aerosols in multiphase and heterogeneous chemistry.

In this study, we used STXM/NEXAFS spectroscopy combined with an environmental cell to image and quantify the chemical morphology and water uptake behavior of individual submicron particles collected in an urban pollution atmosphere. Results show that most organic compounds were internally mixed with inorganic and/or soot, generally presenting a core-shell structure with an inorganic core and an organic shell. Internally mixed particles composed of organic carbon and inorganic matter dominated the particle population by $73.1 \pm 7.4\%$. At 86%RH, 41.6% of the particles took up water, with OCIn particles making up 76.8% of these hygroscopic particles. The relatively low hygroscopicity of bulk aerosols during the sampling period ($\kappa = 0.25 \pm 0.01$) helps to explain the reason why only less than half of the particles took up water. Besides, the majority of particles still showed a heterogeneous core-shell morphology under humid conditions.

This study directly displays the dominant chemical morphology (i.e., core-shell structure) and hygroscopic behavior of individual submicron urban aerosol particles at the microscale. The uptake coefficient onto aerosol particles with different phase states exhibit different patterns as the relative humidity changes (Wang and Lu, 2016). For aqueous particles, the uptake coefficient is closely related to RH (Wang and Lu, 2016). Specifically, when RH is lower than the DRH of the inorganic component, the uptake coefficient increased with the increasing RH. When RH is higher than DRH, the uptake coefficient remains constant. For solid particles, the relationship between the uptake coefficient and RH usually depends on particle species (Wang and Lu, 2016). Results highlight the importance of taking the core-shell structure into consideration for estimating the uptake coefficient and investigating heterogeneous reactions at different humidity, which can improve our comprehension of atmospheric processes of secondary aerosols in typical urban pollution atmospheres.

Moreover, previous studies found that the reactive uptake coefficients of $N_2O_5$ on aqueous sulfuric acid solutions coated with different kinds of organics vary (Cosman and Bertram, 2018; Cosman et al., 2008). The reactive uptake coefficient decreased dramatically for straight-chain surfactants

(1-hexadecanol, 1-octadecanol, and stearic acid) by a factor of 17 – 61 depending on the surfactant type.
While, the presence of branched surfactant phytanic acid didn't show obvious effect on the reactive
uptake coefficient compared to the uncoated solution. These results underlines that the significant
impact of organic species on the reactive uptake coefficient. Therefore, on the basis of the high spectral
energy resolution of STXM/NEXAFS, it is instrumental to conduct research on the effect of organic
molecules and functional groups on heterogeneous reactions in future studies.

ASSOCIATED CONTENT

*Data availability.* The data presented in this article can be accessed through the corresponding author Zhijun Wu via E-mail (zhijunwu@pku.edu.cn).

*Author contributions.* YSZ, PAA, BBW, and JD measured the individual particle sample by STXM/NEXAFS. YSZ, ZJW, YZ, YLG, QC, and SYC carried out the field observation and obtained data. RQM and PAA processed and analyzed data. All authors discussed the results and contributed to the writing of this paper. RQM prepared the manuscript. ZJW, PAA, JC, XRK, MA, and MH further modified and improved the manuscript.

*Competing Interests.* The authors declare that they have no conflict of interest.

*Acknowledgements.* We gratefully acknowledge the Swiss Light Source (SLS) for providing a platform for sample measurements. We also thank Benjamin Watts for helping us dealing with technical problems about STXM/NEXAFS.

*Financial support.* This work has been supported by National Natural Science Foundation of China, International (Regional) Cooperation and Communication Project (NSFC-STINT, China and Sweden; grant No. 42011530121), NSFC (No. 41775133), and the SNSF Swiss Postdoctoral Fellowships (SPF, grant TMPFP2_209830).

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
