# Peer review of "Direct observation of core-shell structure and water"

_EGUsphere, 2025_

## Author Comment (AC1)

**Responses to the reviewers**

We've made every effort to respond to reviewers' questions point to point, and revised our manuscript and the Supplementary Information (SI) according to their comments. For clarity, reviewers' comments are shown in *black italic font*. The response is shown in black normal font. The revised text in the manuscript and/or the SI is shown in **bold blue font**.

**Reviewer: 2**

**General comments:**

This study investigates individual particles from an urban environment using single-particle X-ray microscopy/spectroscopy techniques (STXM/NEXAFS). The authors utilize an environmental cell to explore water uptake potential and report that a significant fraction of the particles are internally mixed with organic and inorganic components and less than half of total particles took up water at 86% relative humidity. Major fractions of these particles exhibit core-shell morphology under both dry and humidified conditions. This is an interesting study, and the use of an environmental cell for water uptake analysis provides critical data for understanding how particle composition influences hygroscopic behavior at the functional group level. I have a few suggestions and questions that may help improve the clarity and impact of the manuscript.

**Specific Comments:**

1. Why were the water uptake experiments performed only up to 86% relative humidity? Was this due to instrumental or operational limitations?

[Response]: Thanks for your questions! The relative humidity (RH) range of STXM environmental cell at SLS used in this study is up to more than 90% (Huthwelker et al., 2010). One of the reasons why we chose 86%RH as humid conditions is that the deliquescence relative humidity (DRH) of commonly presented inorganics is less than 86% (for example, 78-82% for ammonium sulfate, 73-77% for sodium chloride, and 60-66% for ammonium nitrate; Peng et al., 2022). The other reason is for comparing our results with the published work under similar experimental conditions (Pöhlker et al., 2014; Piens et al., 2016).

2. From the images, it appears that OC particles did not uptake water at 86% RH, whereas in some cases, the IN component of OCIN mixed particles exhibited morphological changes. Can the authors expand the discussion on this observation?

[Response]: Thanks for your valuable question! To quantitatively explain the reason why some individual particles took up water while some did not, we tried to take the kappa ( $\kappa$ )-Köhler theory into account to reproduce the observation results. The  $\kappa$ -Köhler equation is as follows (Petters and Kreidenweis, 2007),

$$S(D) = \frac{D^3 - D_d^3}{D^3 - D_d^3 (1 - \kappa)} \exp(\frac{4\sigma_{s/a}M_w}{RT\rho_w D})$$
 Eq. R1

where S is the saturation ratio over an aqueous solution droplet, D is the diameter of the droplet,  $D_d$  is the dry diameter,  $\kappa$  is a single hygroscopicity parameter,  $\sigma_{s/a}$  is the surface tension of the solution/air interface,  $M_w$  is the molecular weight of water, R is the universal gas constant, T is the temperature, and  $\rho_w$  is the density of water.

For gaining the value of the single hygroscopicity parameter ( $\kappa$ ), the simple mixing rule based on the Zdanovskii-Stokes-Robinson (ZSR) assumption is used as follows (Stokes and Robinson, 1966):

$$\kappa = \sum_{i} \varepsilon_{i} \kappa_{i}$$
 Eq. R2

$$\varepsilon_i = \frac{V_{si}}{V_s}$$
 Eq. R3

where i is a component in a multicomponent system (multiple solutes + water),  $\varepsilon_i$  is the dry volume fraction for each component,  $\kappa_i$  is the hygroscopicity parameter of each component,  $V_{si}$  is the volume of each component within a dry particle, and  $V_s$  is the total volume of a dry particle.

However, we acknowledge the inherent challenges in obtaining three-dimensional volume information for each component within individual particles using two-dimensional STXM techniques. This limitation results in a lack of detailed hygroscopicity parameter information for each of the individual particles.

Although we cannot get the  $\kappa$  value for each individual particle, we speculate that the reason why organic carbon (OC) particles didn't take up water under humid conditions while some internally mixed particles composed of organic carbon and inorganic matter (OCIn) did is that the hygroscopic capacity of different components varies. As we mentioned in the manuscript, the  $\kappa$  of ammonium nitrate, ammonium sulfate, ammonium hydrogen sulfate, POA, and SOA is 0.58, 0.48, 0.56, 0, and 0.1, respectively (Wu et al., 2016). Therefore, inorganic matter tend to contribute to water uptake and the morphological change of individual particles.

3. I suggest reorganizing Figures 2 and 3. Instead of displaying particles under dry and humid conditions in separate figures, consider presenting side-by-side comparisons of the same particles before and after humidification. This would help readers more clearly visualize the morphological changes due to water uptake.

[Response]: Thanks for your suggestion! Due to the framework of our manuscript, we chose to display individual particle maps separately under dry and humid conditions. We also take your suggestion into consideration. For comparison, we put together the particle chemical maps of the same region of interest under dry and humid conditions. Please see as follows.

[Revise]: Line 319-320 in the manuscript: "For comparison, the one-to-one particle chemical maps of the same region of interest under both dry and humid conditions can be found in Fig. S9."

**Line 140-144 in the SI:**

Fig. S9: Chemical maps of individual particles in 12 regions of interest under dry and humid conditions. Green, cyan, and red color represent dominant components of organic carbon (OC), inorganic matter (In), and elemental carbon (EC), respectively. The scale bar in the upper left image represents 2  $\mu$ m and applies to all images."

4. The study would benefit from an analysis of the mass growth of individual particles. Did the authors attempt to estimate the mass growth factor using oxygen maps data?

[Response]: Thanks for your valuable suggestion! We also thought about quantifying water uptake of individual particles. Based on the method described in Piens et al. (2016), for calculating the particle mass growth factor, we need to obtain the dry total mass of individual particles and the condensed water mass. The atomic fractions of elements within individual particles are necessary for estimating the dry total mass of particles. However, there is a lack of information regarding atomic fractions obtained using SEM/EDX in our study. Therefore, we failed to get the mass growth factor of particles. We will take it into account in our future research.

5. The conclusion section could be strengthened. For example, what are the implications of observing core-shell morphology under dry conditions? Similarly, how might the presence of such morphologies at 86% RH affect atmospheric processes?

[Response]: Thanks for your comment! We further organized and improved the Conclusions and Implications section. The content is as follows.

[Revise]: Line 428-438 in the manuscript: "This study directly displays the dominant chemical morphology (i.e., core-shell structure) and hygroscopic behavior of individual submicron urban aerosol particles at the microscale. The uptake coefficient onto aerosol particles with different phase states exhibit different patterns as the relative humidity changes (Wang and Lu, 2016). For aqueous particles, the uptake coefficient is closely related to RH (Wang and Lu, 2016). Specifically, when RH is lower than the DRH of the inorganic component, the uptake coefficient increased with the increasing RH. When RH is higher than DRH, the uptake coefficient remains constant. For solid particles, the relationship between the uptake coefficient and RH usually depends on particle species (Wang and Lu, 2016). Results highlight the importance of taking the core-shell structure into consideration for estimating the uptake coefficient and investigating heterogeneous reactions at different humidities, which can improve our comprehension of atmospheric processes of secondary aerosols in typical urban pollution atmospheres."

**References**

- Huthwelker, T., Zelenay, V., Birrer, M., Krepelova, A., Raabe, J., Tzvetkov, G., Vernooij, M. G. C., and Ammann, M.: An in situ cell to study phase transitions in individual aerosol particles on a substrate using scanning transmission x-ray microspectroscopy, Rev. Sci. Instrum., 81, 113706, DOI: 10.1063/1.3494604, 2010.
- Peng, C., Chen, L. X. D., and Tang, M. J.: A database for deliquescence and efflorescence relative humidities of compounds with atmospheric relevance, Fundam. Res., 2, 578–587, DOI: 10.1016/j.fmre.2021.11.021, 2022.
- Petters, M. D. and Kreidenweis, S. M.: A single parameter representation of hygroscopic growth and cloud condensation nucleus activity, Atmos. Chem. Phys., 7, 1961–1971, DOI:10.5194/acp-7-1961-2007, 2007.
- Piens, D. S., Kelly, S. T., Harder, T. H., Petters, M. D., O'Brien, R. E., Wang, B. B., Teske, K., Dowell, P., Laskin, A., and Gilles, M. K.: Measuring Mass-Based Hygroscopicity of Atmospheric Particles through in Situ Imaging, Environ. Sci. Technol., 50, 5172–5180, DOI: 10.1021/acs.est.6b00793, 2016.
- Pöhlker, C., Saturno, J., Krüger, M. L., Förster, J. D., Weigand, M., Wiedemann, K. T., Bechtel, M., Artaxo, P., and Andreae, M. O.: Efflorescence upon humidification? X-ray microspectroscopic in situ observation of changes in aerosol microstructure and phase state upon hydration, Geophys. Res. Lett., 41, 3681–3689, DOI: 10.1002/2014gl059409, 2014.
- Stokes, R. H. and Robinson, R. A.: Interactions in aqueous nonelectrolyte solutions. I. Solute-solvent equilibria, J. Phys. Chem., 70, 2126–2131, DOI: 10.1021/j100879a010, 1966.
- Wu, Z. J., Zheng, J., Shang, D. J., Du, Z. F., Wu, Y. S., Zeng, L. M., Wiedensohler, A., and Hu, M: Particle hygroscopicity and its link to chemical composition in the urban atmosphere of Beijing, China, during summertime, Atmos. Chem. Phys., 16, 1123–1138, DOI: 10.5194/acp-16-1123-2016, 2016.

---

## Author Comment (AC2)

**Responses to the reviewers**

We've made every effort to respond to reviewers' questions point to point, and revised our manuscript and the Supplementary Information (SI) according to their comments. For clarity, reviewers' comments are shown in *black italic font*. The response is shown in black normal font. The revised text in the manuscript and/or the SI is shown in **bold blue font**.

**Reviewer: 1**

General comments: This manuscript presents results from an analysis of an urban aerosol sample collected on a substrate and analyzed using STXM/NEXAFS in a cell that enables RH control. The composition of the particles was characterized along with the changes at high RH. A large fraction of the particles displayed core/shell morphology with an inorganic core and an organic coating. Many particles also had signal for black carbon or soot. Many of the particles took up water and generally the particles became smoother at higher RH. Some information collected at the same time on the submicron aerosol population is also presented. Overall, this is an interesting measurement report on these particles. These types of studies are challenging to do, and I appreciate the care that was taken in terms of the loading on the substrates to enable analysis of ambient urban particles. However, there are many places where broad general statements are made that could be more specific. There are also some locations where more information is needed to clarify the study or the conclusions that are being drawn. After addressing these concerns, I think this manuscript will be of interest to the readers of EGU Sphere and I recommend acceptance.

**Specific comments:**

1. The introduction lacks information on the location for the sample collection. Please include that.

[Response]: Thanks for your comment! The relevant descriptions were added as follows. More detailed information about sampling site could be found in the Method Sec. (Line 118-121: To study the physicochemical properties of ambient particles, samples were collected during a pollution episode at the Peking University Urban Atmosphere Environment Monitoring Station (PKUERS, 39°59′21″N, 116°18′25″E) in Beijing, China. More details about the measurement site can be found in our previous studies (Tang et al., 2021; Wu et al., 2007).)

[Revise]: Line 110-111 in the manuscript: "The ambient samples were collected at an urban site in Beijing of North China Plain during a pollution episode."

2. Both the introduction and the conclusions have general statements about a need for physicochemical properties for aerosol particles. One example is given for why this is needed to understand reactive uptake/multiphase processes ( $N_2O_5$ ). This is a big field, and I think a stronger and more thorough background on what is known and what is not known in urban multiphase chemistry would really help this article. Right now, I don't have a good understanding of why the type of information that this study gives is helpful except the general statement that core/shell morphology is important. What are the knowledge gaps that this study is filling?

[Response]: Thanks for your constructive comments! In brief, the novelty of our study is that we first explored the chemical morphology as well as the water uptake and hygroscopic behaviors of individual submicron urban aerosol particles using STXM/NEXAFS spectroscopy. We added relevant descriptions of the background and the knowledge gap of studies on physicochemical properties of urban aerosols in China. Please see as follows.

[Revise]: Line 100-110 in the manuscript: "In recent years, the air quality in China has improved notably due to the implementation of a series of strict pollution mitigation measures. These improvements are attributed to decreasing primary emissions, while the contributions of secondary species to particle mass have become more significant (Lei et al., 2021; Wang et al., 2019). To elucidate the causes and mechanisms of pollution episodes in China, numerous research has been carried out on the pollution characteristics (Gao et al., 2015; Gao et al., 2018; Guo et al., 2014; Huang et al., 2014; Liu et al., 2018; Sun et al., 2013; Wang et al., 2014; Zhao et al., 2013) and physicochemical properties (Gao and Anderson, 2001; Li et al., 2017; Shen et al., 2019; Song et al., 2022) of ambient aerosols. However, there is still a lack of studies on direct observation of the chemical morphology and hygroscopic behaviors of secondary urban aerosols at the single particle level. This knowledge gap hinders our understanding of the role of secondary aerosols as reaction vessels in heterogeneous reactions."

3. For the paragraph starting on line 66, it is stated that extensive research has been conducted on physiochemical properties of bulk aerosol. (1) I'm a little confused by this statement because these are online techniques and some of the instruments can do single particle analysis. I understand the distinction you are making between these measurements and your single-particle imaging analysis. But I would recommend rephrasing and being a little more precise about what is being measured. (2) The methods you mentioned also have a big strength in statistics that STXM/NEXAFS lacks, and it would be good to present a more balanced comparison between all the different methods.

[Response]: Thanks for your comments!

(1) To the best of our knowledge, Humidified Tandem Differential Mobility Analyzer (H-TDMA), Aerosol Mass Spectrometer (AMS), and Soot Particle Aerosol Mass Spectrometer (SP-AMS) ("SP" herein refers to "soot particle", rather than "single particle") all target particle populations.

(2) For single particle analysis, although commonly used individual particle techniques (such as TEM and SEM) have advantages from the statistics aspect, they have shortcomings that STXM/NEXAFS could make up well. The advantages and disadvantages of different kinds of individual particle techniques are shown in Fig. R1. For example, Transmission electron microscopy (TEM) offers high spatial resolution, while it is not well-suited for analyzing organic species (especially compounds containing nitrogen), which are dominant in ambient aerosols (Moffet et al., 2011; Shao et al, 2022). Scanning electron microscopy (SEM) can obtain the three-dimensional morphological characteristics of particles with high counting statistics, but it is limited to get information only on the particle surface and quantitative elemental analysis of heavy elements (Z > 11) (Moffet et al., 2011; Shao et al, 2022).

Compared to TEM and SEM, the main advantages of STXM/NEXAFS spectroscopy includes: (1) chemical specificity, (2) higher spectral energy resolution, (3) lower excitation energy (100 - 2000 eV for STXM versus 50 - 200 keV for TEM and SEM) and corresponding reduced radiation exposure, and (4) no ultrahigh vacuum requirements (Moffet et al., 2011).

In short, STXM/NEXAFS spectroscopy provides an enhanced chemical sensitivity for obtaining specific organic chemical bonds, functional groups, and speciation information. Therefore, it is an appropriate technique for exploring not only the particle morphology but also the detailed chemical components (especially for organic species) of individual ambient particles. Please see Line 78-88 in the manuscript.

Fig. R1: Analysis of capabilities (x-axis) of imaging techniques (y-axis) used to analyze atmospheric aerosols (Moffet et al., 2011). The gray scale corresponds to analysis capability which is classified as strong, medium, or weak. Acronyms are as follows: TEM/EELS/EDX/SAED: transmission electron microscopy/electron energy

loss spectroscopy/energy dispersive x-ray/selected area electron diffraction, SEM: scanning electron microscopy, FTIR: Fourier transform infrared spectroscopy, XRF/XAS: x-ray fluorescence/x-ray absorption spectroscopy, PIXE: proton-induced x-ray emission, and TOF-SIMS: time-of-flight secondary ion mass spectrometry.

[Revise]: Line 78-88 in the manuscript: "It is a robust technique for obtaining chemical morphology information of numerous individual particles with high spectral energy resolution and chemical specificity, as it can identify and distinguish various chemical composition at the single particle level within a particle population (Moffet et al., 2011; Shao et al., 2022). The soft X-ray energy range of STXM (100 - 2000 eV for STXM versus 50 - 200 keV for electron microscopy) makes it possible to quantify light elements (such as carbon, nitrogen, and oxygen) with little beam damage (Moffet et al., 2011). In addition, STXM doesn't require ultrahigh vacuum conditions (Moffet et al., 2011). In short, STXM/NEXAFS spectroscopy provides an enhanced chemical sensitivity for obtaining specific organic chemical bonds, functional groups, and speciation information, which has enormous potential in exploring ambient samples under atmospheric relevant conditions, especially submicron-sized particles."

4. On line 80 it is noted "as it can resolve compositional contrast at the single particle level.". I'm not sure what compositional contrast means, please rephrase.

[Response]: Thanks for your comment! When X-ray beam goes through samples, a part of X-rays are absorbed and transmitted X-rays are detected. Amount of absorption depends on photon energy, elemental composition, density, and sample thickness (Moffet et al., 2011). Therefore, the intensity of transmitted X-rays through various chemical composition varies at a certain photon energy, leading to gray scale contract between different chemical composition on individual particle maps (see Figs. S4-S6). That's what "compositional contrast" means. For clarity, we rephrased the sentence. Please see as follows.

[Revise]: Line 78-82 in the manuscript: "It is a robust technique for obtaining chemical morphology information of numerous individual particles with high spectral energy resolution and chemical specificity, as it can identify and distinguish various chemical composition at the single particle level within a particle population (Moffet et al., 2011; Shao et al., 2022)."

5. You note that these samples are frozen on page 5. Do you anticipate any changes during freezing? Have any studies been done to test this for these samples? I know other groups avoid freezing before imaging work.

[Response]: Thanks for your valuable question! We know some researchers sealed and stored their samples at ambient ( $\sim$ 21°C) or relatively low temperature (4°C) and low relative humidity (20-30% or  $\sim$ 50%) (Pöhlker et al., 2014; Mikhailov et al., 2015; Moffet et al., 2016). We chose to store samples filled with nitrogen gas in a freezer at a temperature of -18°C, mainly to avoid the loss of volatile components of

particles and prevent additional exposure to light and moisture. We didn't carry out relevant research on the effect of freezing samples. We are going to find it out in our future study.

6. Were the spectra collected on the same particles as the ones imaged for figures 2 and 3? Was there any evidence for beam damage in these samples?

[Response]: Thanks for your questions! The spectra displayed in Fig. 4 were collected on the same particles as shown in Figs. 2 and 3. The beam damage of STXM has been reported to be little by the published work (Moffet et al., 2011). To confirm it, we obtained several spectra over the same spot under different photon energies at the carbon and nitrogen K-edges. The results are as follows:

Fig. R2: Spectra collected over the same spot at different photon energies at the (A) carbon and (B) nitrogen K-edges.

As displayed in Fig. R2, the optical density at different photon energies of several scans resembles each other at both the carbon and nitrogen K-edges, which indicates no obvious beam damage occurred.

7. For the quantification of the water uptake, was any change in the baseline observed in the oxygen spectra? I'm curious if you can see a thin water layer on the substrate surface and how this changes the cut-off for the particle diameter.

[Response]: Thanks for your questions! We measured samples only at the preand post-edges of the oxygen K-edge, so there is no baseline in the oxygen spectra. We didn't see water layer on the hydrophobic copper substrate under humid conditions. As shown in Fig. 3, the individual particles presents separately and their edges and corresponding particle diameters are easily to be distinguished by pixels.

8. In Figure 1 you show a pie chart for the full pollution episode. How does the pie chart in the period around your sample compare? This can be in the supplemental, but I'm having trouble eyeballing it to compare.

[Response]: Thanks for your suggestion! The pie chart of the average mass fractions of particle chemical composition measured by AMS during the sampling

period of the individual particle sample was added in the SI, and the relevant descriptions were added in the manuscript.

[Revise]: Line 243-247 in the manuscript: "The average mass fractions of chemical composition of NR-PM1 during the sampling period of individual particles could be found in Fig. S3. During this period, the low mass fraction of volatile inorganic species such as nitrate made it suitable for measurements using offline techniques, such as STXM, because the loss of volatile species during storage and measurement processes was minimal."

**Line 114-117 in the SI:**

Fig. S3 Pie chart showing the average mass fractions of chemical composition of non-refractory submicron particles (NR-PM1) during the sampling period of individual particles."

9. In Figure 1 the pie chart for POA vs. SOA is shown. How do these factors vary with time during the campaign? Do you expect to see more POA in your sample from traffic?

[Response]: Thanks for your questions! Time series of primary organic aerosol (POA) and secondary organic aerosol (SOA) are shown as follows.

Fig. R3: Time series of mass concentrations of primary organic aerosol (POA, light green) and secondary organic aerosol (SOA, dark green) during the pollution episode.

As shown in Fig. R3, there are usually two common peaks at the SOA mass concentrations every day, respectively at 12:00-14:00 at noon, and 19:00-23:00 in the evening and at night. Besides, two peaks appeared at the mass concentrations of POA, one is at 12:00-14:00 at noon, the other is at 19:00-20:00 in the evening. The second peak usually shows higher mass concentrations, which may be emitted from motor vehicles during evening rush hours.

10. On line 268 it is stated that "EC/ soot (colored in red), found either near the center or the edge of the individual particles". This is a very general statement and I'm not sure why it is being made. What are the other options for its location in an internally mixed particle except for near the center or the edge?

[Response]: Thanks for your comment! The redundant descriptions were deleted and the content is as follows.

[Revise]: Line 283-284: "Approximately one quarter (24.9%) of the particles contained EC/soot (colored in red). Notably, around 82% of these soot-containing particles had soot located at particle edges."

11. Just below that on line 269 a possible reason for soot on the particle edges is given. Is this the only possibility?

[Response]: Thanks for your question! The reason why soot located at the particle edges may be various, for example, soot may be emitted and condensed onto the particles during the aging processes of aerosols. Here, we have merely elucidated one of the potential reasons presented in the published study. To make it more accurate, we revised the relevant descriptions. Please see as follows.

[Revise]: Line 284-286: "One of the possible reasons is that inorganic species (such as crystals) pushed soot away from the center of the particles during their efflorescence (Moffet et al., 2016)."

12. On line 271 it is noted that the soot showed fractal or compact structures of various sizes. I'm a bit uncomfortable with the statement that these are fractal. I don't think you have the resolution needed to really characterize the soot at that level. I would recommend rephrasing.

[Response]: Thanks for your correction! We removed the relevant descriptions based on your comments.

13. (1) On line 278 it is noted that soot particles with thin coatings have smaller absorption enhancements compared to thick coatings. This is true, but does that apply here? (2) How does the position of the soot within the particle impact this? Would you expect the same type of absorption enhancement for a soot particle on the edge compared to one in the center?

[Response]: Thanks for your comments!

- (1) Particle absorption enhancements are significantly influenced by the thickness of coating, as well as the mixing state and morphology of black carbon (BC) (China et al., 2015; Lee et al., 2019). Therefore, when we discuss about the effect of the coating thickness of particles, we assume that other factors remain constant. Based on the assumption, we think it is reasonable to use the pattern here.
- (2) The locations of BC inclusions within the individual particles also affect the optical properties of particles (Fuller et al., 1999). Specifically, the soot-containing particles with concentrically located BC shows higher specific absorption than those with arbitrarily located BC (Fuller et al., 1999).
- 14. The O/C of the organic is estimated at the bottom of page 11. How does this compare to the O/C measured with the AMS in the same time range?

[Response]: Thanks for your question! We didn't compare the O:C ratio obtained by STXM and LToF-AMS, because they are based on distinct sampling methods and estimation principles. The main differences are as follows:

- (1) Sampling object: STXM measures individual particles, while AMS targets bulk aerosols;
- (2) Particle diameter: Particles collected onto the last stage using a four-stage cascade impactor with the 50% cut-point aerodynamic diameter of 250-nm were used for STXM analysis, while AMS measured the non-refractory chemical composition of submicron particles (NR-PM1);
- (3) Estimation principle: The O:C ratio estimated using STXM data is based on the optical density and the atomic photoabsorption cross section of carbon and oxygen (detailed methods could be found in the SI), which targets all chemical components within individual particles. While, the O:C ratio calculated by AMS is obtained in terms of the relative mass concentrations of oxygen and carbon of organic molecules (Aiken et al., 2007; Aiken et al., 2008), which therefore aims at only organic matter.

Therefore, in our opinion, the O:C ratio calculated based on STXM and AMS data is not comparable.

15. On line 315 it is stated "This phenomenon aligns with a previous study which indicates that the phase transition of phase-separated particles without phase mixing will not cause the redistribution of soot within individual particles..". I'm not sure what this sentence means, and I recommend rephrasing.

[Response]: Thanks for your comment! The relevant descriptions were revised as follows.

[Revise]: Line 331-333 in the manuscript: "A previous study witnessed the redistribution of soot within phase-separated particles only after the phase mixing process occurred (Zhang et al., 2022), which is consistent with the phenomenon observed in our study."

16. For Figure 2 vs. Figure 3, I can see some differences in the particles that are interesting and not discussed in the manuscript. In Figure 3 there are particles in i, v, and vii, that had clear EC/soot in Figure 2 and now lack a clear EC/soot signal in Figure 3 at high RH. Why is this happening? In iii I see a particle that is fully green (org) in Figure 2 but that has some blue inclusions (inorganic) in Figure 3. What is driving these changes and does this say anything about thresholds in the cutoffs for the different components (OC, IN, EC)?

[Response]: Thanks for your constructive questions! During the process of obtaining the chemical maps of individual particles, we used the same thresholds as described in Moffet et al. (2010) and Moffet et al. (2016). Specifically, the criteria are as follows: (1) pixels at 288.6 eV with intensities 3 times below the signal to noise ratio were set to zero; (2) pixels having  $OD_{pre} / OD_{post}

18. On line 359 it is noted that the peak at 288.6 eV is always found at the outer shell of particles. This seems odd to me as I have seen this peak when I have looked at the center of organic particles. Please clarify.

[Response]: Thanks for your valuable comment! Even if a particle presents a core-shell structure with an inorganic-rich core and an organic-rich shell, there can be extra components in both the core and the shell, for example, organic in the core or inorganic in the shell (Gaikwad et al., 2022). The peak appearing at 288.6 eV

represents the characteristic transition of carboxylic carbonyl functional groups, which refers to organic matter (Moffet et al., 2016). Therefore, it may also exist in the particle core. As shown in Fig. 4, the intensity of peaks at different photon energies varies from different regions of individual particles.

19. It is noted that potassium could correspond to biomass burning. Was there any evidence for biomass burning in the AMS data?

[Response]: Thanks for your question! Due to the availability of data, we didn't get information about the mass concentrations of biomass burning tracers, such as levoglucosan, potassium ion, or acetonitrile. Therefore, We inferred that the biomass burning process is a possible cause according to the published papers that showed biomass burning is generally an important pollution source in autumn in Beijing (Xu et al., 2020; Liang et al., 2023).

20. On line 381 it is noted that a peak is characteristic of sulfate-rich particles. Please include a citation for this statement or some standards for comparison.

[Response]: Thanks for your comment! The references were cited in the latter sentence about sulfate-rich particles. For clarity, we shifted them to former one. The content is as follows.

[Revise]: Line 396-400 in the manuscript: "Oxygen K-edge spectra in Fig. 4C exhibited a large peak at 536.9 eV, which is a representative characteristic of sulfate-rich particles (Colberg et al., 2004; Slowik et al., 2011; Mikhailov et al., 2015; Pöhlker et al., 2014), consistent with the result of AMS. A smaller peak was observed at 532.5 eV, confirming the presence of ketone, aldehyde, or carboxyl functionalities (Moffet et al.,2011), which aligns with the results from C K-edge spectra."

21. In the first sentence of the conclusions, I don't understand what is being referred to by the word "which". Please clarify this sentence.

[Response]: Thanks for your correction! In this sentence, "which" refers to "heterogeneous reactive uptake of gaseous molecules". The relevant descriptions were revised as follows.

[Revise]: Line 410-412 in the manuscript: "Particles in the atmosphere usually act as reaction vessels for heterogeneous reactive uptake of gaseous molecules, and heterogeneous processes play an important part in gas-particle partitioning and secondary aerosol formation (Abbatt et al., 2012; Davidovits et al., 2011; Kolb et al., 2010)."

22. The last couple of statements are very broad and the last sentence is not clear to me what future studies are being proposed. I recommend being more specific here.

[Response]: Thanks for your comment! We organized and improved the last paragraph. Please see as follows.

[Revise]: Line 439-447 in the manuscript: "Moreover, previous studies found that the reactive uptake coefficients of N2O5 on aqueous sulfuric acid solutions coated with different kinds of organics vary (Cosman and Bertram, 2018; Cosman et al., 2008). The reactive uptake coefficient decreased dramatically for straight-chain surfactants (1-hexadecanol, 1-octadecanol, and stearic acid) by a factor of 17 - 61 depending on the surfactant type. While, the presence of branched surfactant phytanic acid didn't show a obvious effect on the reactive uptake coefficient compared to the uncoated solution. These results underlines that the significant impact of organic species on the reactive uptake coefficient. Therefore, on the basis of the high spectral energy resolution of STXM/NEXAFS, it is instrumental to conduct research on the effect of organic molecules and functional groups on heterogeneous reactions in future studies."

**References**

- Aiken, A. C., DeCarlo, P. F., and Jimenez, J. L.: Elemental Analysis of Organic Species with Electron Ionization High-Resolution Mass Spectrometry, Anal. Chem., 79, 8350–8358, DOI:10.1021/ac071150w, 2007.
- Aiken, A. C., DeCarlo, P. F., Kroll, J. H., Worsnop, D. R., Huffman, J. A., Docherty, K., Ulbrich, I. M., Mohr, C., Kimmel, J. R., Sueper, D., Sun, Y., Zhang, Q., Trimborn, A., Northway, M., Ziemann, P. J., Canagaratna, M. R., Onasch, T. B., Alfarra, M. R., Prevot, A. S. H., Dommen, J., Duplissy, J., Metzger, A., Baltensperger, U., and Jiménez, J. L.: O/C and OM/OC Ratios of Primary, Secondary, and Ambient Organic Aerosols with a High Resolution Time-of-Flight Aerosol Mass Spectrometer, Environ. Sci. Technol., 42, 4478–4485, 2008.
- China, S., Scarnato, B., Owen, R. C., Zhang, B., Ampadu, M. T., Kumar, S., Dzepina, K., Dziobak, M. P.; Fialho, P., Perlinger, J. A., Hueber, J., Helmig, D., Mazzoleni, L. R., and Mazzoleni, C.: Morphology and mixing state of aged soot particles at a remote marine free troposphere site: Implications for optical properties, Geophys. Res. Lett., 42, 1243–1250, DOI: 10.1002/2014gl062404, 2015.
- Fuller, K. A., Malm, W. C., and Kreidenweis, S. M.: Effects of mixing on extinction by carbonaceous particles, J. Geophys. Res.: Atmos., 104, 15941–15954, DOI: 10.1029/1998JD100069, 1999.
- Gaikwad, S., Jeong, R., Kim, D., Lee, K., Jang, K.-S., Kim, C., and Song, M.: Microscopic observation of a liquid-liquid-(semi)solid phase in polluted PM2.5, Front Env. Sci. Switz, 10, DOI: 10.3389/fenvs.2022.947924, 2022.
- Lee, A. K. Y., Rivellini, L. H., Chen, C. L., Liu, J., Price, D. J., Betha, R., Russell, L. M., Zhang, X. L., and Cappa, C. D: Influences of Primary Emission and Secondary Coating Formation on the Particle Diversity and Mixing State of Black Carbon Particles, Environ. Sci. Technol., 53, 9429–9438, DOI: 10.1021/acs.est.9b03064, 2019.
- Liang, L., Du, Z., Engling, G., Liu, X., Xu, W., Liu, C., Cheng, Y., Ji, D., Zhang, G., and Sun, J.: Improved biomass burning pollution in Beijing from 2011 to 2018, Atmos. Environ., 310, 119969, DOI: 10.1016/j.atmosenv.2023.119969, 2023.
- Mikhailov, E. F., Mironov, G. N., Pöhlker, C., Chi, X., Krüger, M. L., Shiraiwa, M., Förster, J. D., Pöschl, U., Vlasenko, S. S., Ryshkevich, T. I., Weigand, M., Kilcoyne, A. L. D., and Andreae, M. O.: Chemical composition, microstructure, and hygroscopic properties of aerosol particles at the Zotino Tall Tower Observatory (ZOTTO), Siberia, during a summer campaign, Atmos. Chem. Phys., 15, 8847–8869, DOI: 10.5194/acp-15-8847-2015, 2015.
- Moffet, R. C., Henn, T., Laskin, A., and Gilles, M. K.: Automated Chemical Analysis of Internally Mixed Aerosol Particles Using X-ray Spectromicroscopy at the Carbon K-Edge, Anal. Chem., 82, 7906–7914, DOI: 10.1021/ac1012909, 2010.
- Moffet, R. C., O'Brien, R. E., Alpert, P. A., Kelly, S. T., Pham, D. Q., Gilles, M. K., Knopf, D. A., and Laskin, A.: Morphology and mixing of black carbon particles collected in central

- California during the CARES field study, Atmos. Chem. Phys., 16, 14515–14525, DOI: 10.5194/acp-16-14515-2016, 2016.
- Moffet, R. C., Tivanski, A. V., and Gilles, M. K.: Scanning Transmission X-ray Microscopy Applications in Atmospheric Aerosol Research, Fundamentals and Applications in Aerosol Spectroscopy, edited by: Signorell, R., and Reid, J. P., CRC Press, the U.S., 419–462, ISBN: 9781420085617, 2011.
- Pöhlker, C., Saturno, J., Krüger, M. L., Förster, J. D., Weigand, M., Wiedemann, K. T., Bechtel, M., Artaxo, P., and Andreae, M. O.: Efflorescence upon humidification? X-ray microspectroscopic in situ observation of changes in aerosol microstructure and phase state upon hydration, Geophys. Res. Lett., 41, 3681–3689, DOI: 10.1002/2014gl059409, 2014.
- Shao, L. Y., Liu, P. J., Jones, T., Yang, S. S., Wang, W. H., Zhang, D. Z., Li, Y. W., Yang, C.-X., Xing, J. P., Hou, C., Zhang, M. Y., Feng, X. L., Li, W. J., and BéruBé, K.: A review of atmospheric individual particle analyses: Methodologies and applications in environmental research, Gondwana Research, 347–369, DOI: 10.1016/j.gr.2022.01.007, 2022.
- Wu, Z. J., Zheng, J., Shang, D. J., Du, Z. F., Wu, Y. S., Zeng, L. M., Wiedensohler, A., and Hu, M: Particle hygroscopicity and its link to chemical composition in the urban atmosphere of Beijing, China, during summertime, Atmos. Chem. Phys., 16, 1123–1138, DOI: 10.5194/acp-16-1123-2016, 2016.
- Xu, S., Ren, L., Lang, Y., Hou, S., Ren, H., Wei, L., Wu, L., Deng, J., Hu, W., Pan, X., Sun, Y., Wang, Z., Su, H., Cheng, Y., and Fu, P.: Molecular markers of biomass burning and primary biological aerosols in urban Beijing: size distribution and seasonal variation, Atmos. Chem. Phys., 20, 3623–3644, DOI: 10.5194/acp-20-3623-2020, 2020.

---

## Author Response (AR2)

***Responses to the reviewer***

Thanks for the reviewer's helpful comments and suggestions again. We made every effort to respond to the reviewer's questions point to point, and improved our manuscript and supplementary information (SI) accordingly. For clarity, the reviewer's comments are shown in *black italic font*. The response is shown in black normal font. The revised text in the manuscript and/or the SI is shown in **bold blue font**.

*General comments: Thank you for your work on these revisions. You have addressed many of my concerns, but there are a few notes I would like to follow up on. I gently recommend that more effort be placed in making changes in the text of the manuscript or in the supplemental to address comments or questions that a reviewer raises.*

*Specific comments:*

*1. I requested information on freezing for these samples and this was provided in the response, but no changes were made in the document. Please add some notes either to the manuscript or the supplement to provide context for the possible effect of freezing on the samples.*

[Response]: Thanks for your comment! We searched for more research on freezing effects on field samples during storage. The relevant descriptions were added as follows.

**[Revise]: Line 130-136 in the manuscript: "Previous results indicate that the chemical composition of organic aerosols (especially secondary organic aerosols, SOA) and the mass concentrations of black carbon (BC) both remained stable for several weeks under low-temperature storage conditions (–20°C and –80°C for organic aerosols, and 2°C, 4°C, and 5°C for BC); while significant changes occurred over time when samples were stored at room temperature even just for a few days (Mori et al., 2016; Mori et al., 2019; Resch et al., 2023; Ueda et al., 2025; Wendl et al., 2014)."**

*2. I requested information about the possible beam damage in the samples. I am happy to see that minimal signal changes were observed, please add this information either to the manuscript or the supplement as it will be helpful for future studies in this area.*

[Response]: Thanks for your suggestion! We added information about beam damage in the manuscript and the SI. Please see as follows.

**[Revise]: Line 177-179 in the manuscript: "Optical density detected over the same spot at different photon energies at the carbon and nitrogen K-edges was displayed in Fig. S1, and less beam damage during the experiment was confirmed."**

**Line 107-112 in the SI:**

[Figure]

[Figure]

"

**Figure S 1: Spectra collected over the same spot at different photon energies at the (A) carbon and (B) nitrogen K-edges.**

**As displayed in Fig. S1, optical density at different photon energies of several scans resembles each other at both the carbon and nitrogen K-edges, which indicates no obvious beam damage occurred."**

*3. In a comment I requested information on the absorption enhancements and soot location. The response to reviewer comments makes some good statements, but the underlying question is not addressed, and no changes were made to the manuscript. Please consider revising this portion of the manuscript. To clarify my concerns, I will pose the following problem:*

*if a particle has the black carbon on the edge of the particle, but the particle lands such that the soot is centrally located in the impacted/collected particle, you cannot tell that this is a particle with soot on the edge vs. soot that is at the center. This problem adds some uncertainty in all estimates of morphology from STXM data, and it should be clearly stated so that this and future data sets are not over-interpreted in terms of absorption enhancements based on organic/BC morphology.*

[Response]: Thanks for your valuable comments! We added some descriptions in the manuscript. Please see as follows.

**[Revise]: Line 294-298 in the manuscript: "While, it should be noted that particle deformation may occur during the particle collection process due to the high particle impaction velocity of the impactor (O'Brien et al., 2014). Therefore, the distribution of chemical components within individual particles displayed in the images may differ from that of aerosol particles in ambient atmospheres."**

*4. I asked for a comparison of the STXM O/C with the AMS O/C. In the response, several reasons for why these might be different are provided and it is stated that they are not comparable. I understand the limitations, but please provide the O/C values from the AMS data sets for these time periods either in the main text or the supplemental. You are welcome to clarify the limitations of the comparisons, but*

*future studies may want to make those comparisons, and the data should be provided here to enable that. For this comparison, I would also argue that your third statement in the response does not make sense compared to what is stated in the manuscript. In the manuscript, the O/C from STXM for the organic dominated particles is used to argue that LLPS is possible. Why would the AMS data not be comparable to this if you are focusing on organic dominated particles for the STXM O/C calculations?*

[Response]: Thanks for your comments! We added data set in the SI and the relevant descriptions in the manuscript. Please see as follows.

**[Revise]: Line 313-319 in the manuscript: "For comparison, the value of the O:C ratio by AMS during the individual particle collection period is also calculated (0.60 ± 0.01), and the data set was displayed in the SI. The difference between the O:C ratio results by STXM and LToF-AMS may be due to the reasons as follows: (1) STXM measures individual particles, while AMS targets bulk aerosols; (2) Particles collected onto the last stage using a four-stage cascade impactor with the 50% cut-point aerodynamic diameter of 250-nm were used for STXM analysis, while AMS measured the non-refractory chemical composition of submicron particles."**

**Line 92-94 in the SI:**

**"Table S 2 The data set of the oxygen-to-carbon (O:C) ratio based on AMS data using "Improved-Ambient" Methods (Canagaratna et al., 2015) during the individual particle collection period."**

| Date and Time | O:C |
|---|---|
| **2019/10/1 17:05** | **0.61** |
| **2019/10/1 17:06** | **0.61** |
| **2019/10/1 17:07** | **0.60** |
| **2019/10/1 17:08** | **0.59** |
| **2019/10/1 17:09** | **0.58** |

*5. I asked about the note that is made in the manuscript that the peak at 288.6 eV is always found at the outer shell of the particles. This statement is now on line 375 of the document with track changes. The response to reviewer comments note that this can be observed in the particles, but no corrections were made to the text in the manuscript. Please correct the text in this sentence.*

[Response]: Thanks for your comment! We added the relevant descriptions in the manuscript. Please see as follows.

**[Revise]: Line 391-397 in the manuscript: "The peak appearing at 288.6 eV represents the characteristic transition of carboxylic carbonyl functional groups (C 1s→π*$_{R(C*=O)OH}$), which refers to organic matter and is generally dominant in the outer shell of particles (Moffet et al., 2016; Prather et al., 2013). One should note that there could be extra components in both core and shell in a**

phase-separated particle with an inorganic-rich core and an organic-rich shell, for example, organic in core or inorganic in shell (Gaikwad et al., 2022). Therefore, the 288.6-eV peak may also be observed in a particle core with relatively low peak intensity."